# The Role of RNA in DNA Breaks, Repair and Chromosomal Rearrangements

**DOI:** 10.3390/biom11040550

**Published:** 2021-04-09

**Authors:** Matvey Mikhailovich Murashko, Ekaterina Mikhailovna Stasevich, Anton Markovich Schwartz, Dmitriy Vladimirovich Kuprash, Aksinya Nicolaevna Uvarova, Denis Eriksonovich Demin

**Affiliations:** 1Engelhardt Institute of Molecular Biology, Russian Academy of Sciences, 119991 Moscow, Russia; murashko.mm@phystech.edu (M.M.M.); stasevich.em@phystech.edu (E.M.S.); shvarts@eimb.ru (A.M.S.); kuprash@eimb.ru (D.V.K.); uan@eimb.ru (A.N.U.); 2Moscow Institute of Physics and Technology, Department of Molecular and Biological Physics, 141701 Moscow, Russia

**Keywords:** RNA, chromosomal rearrangements, double-strand break, DNA repair

## Abstract

Incorrect reparation of DNA double-strand breaks (DSB) leading to chromosomal rearrangements is one of oncogenesis’s primary causes. Recently published data elucidate the key role of various types of RNA in DSB formation, recognition and repair. With growing interest in RNA biology, increasing RNAs are classified as crucial at the different stages of the main pathways of DSB repair in eukaryotic cells: nonhomologous end joining (NHEJ) and homology-directed repair (HDR). Gene mutations or variation in expression levels of such RNAs can lead to local DNA repair defects, increasing the chromosome aberration frequency. Moreover, it was demonstrated that some RNAs could stimulate long-range chromosomal rearrangements. In this review, we discuss recent evidence demonstrating the role of various RNAs in DSB formation and repair. We also consider how RNA may mediate certain chromosomal rearrangements in a sequence-specific manner.

## 1. Introduction

Well-established and newly released data increasingly suggest that various DNA processes in the nucleus of the eukaryotic cells are associated with certain types of RNA. It has been shown that various RNAs play key roles in the processes of replication, repair, telomere lengthening, regulation of the structure of chromosomes and formation of dynamic chromatin structures [1,2,3,4]. The formation of chromosomal rearrangements is also linked to the activity of several RNAs [5,6,7,8].

Chromosomal rearrangements usually appear due to incorrect reparation of double-strand breaks (DSBs) [9]. The damage can be done by ionizing and UV radiation, chemical carcinogens, viruses or mobile genetic elements. The probability of such breaks is increased in the proximity of DNA:RNA hybrids, or so-called R-loops [10]. They separate DNA chains and increase their vulnerability to carcinogens [11]. Such structures can occur during transcription and interaction of DNA with regulatory RNAs [12]. Moreover, the recognition and repair of DSBs are associated with the transcription apparatus, and in some cases, these processes recruit RNA synthesized from the sequence in which the break occurred [12].

A number of different types of noncoding RNAs synthesized from other regions of the genome participate in regulating the expression of repair factors, their organization into complexes, and attraction of these complexes to the site of rupture [13]. Changes in the expression of such RNAs may significantly influence the probability of correct reparation of DSB. For example, increased expression of microRNAs (miRNA) miR-1255b, miR-193b and miR-148b promote repair by the less precise NHEJ mechanism by suppressing the expression of homologous recombination (HR) factors BRCA1, BRCA2, and RAD51 [14]. Furthermore, it has been demonstrated that certain RNAs can stimulate long-range chromosomal rearrangements. In ciliates, noncoding RNAs have a well-studied role in guiding genome rearrangement [15]. Recently, it was shown that individual RNAs could stimulate specific chromosomal rearrangements in human cells [6]. In addition to endogenous RNAs, transcriptional events that aggravate chromosomal mutations can be of viral origin. In particular, retroviruses can inactivate tumor-suppressor genes, activate cellular oncogenes or bring their own oncogenes, such as v-Src from Rous sarcoma virus or Tax from Human T-cell lymphotropic virus type 1 [16,17]. In this review, we examine in detail the role of different types of RNA in the regulation of the mechanisms of DSB formation, recognition and reparation, deviation in which may lead to a consequent chromosomal rearrangement formation, and we also discuss the significance of these RNAs for the process of oncogenesis.

## 2. Formation of Double-Strand Breaks Is Provoked by RNA

As indicated above, the first step in the formation of chromosomal rearrangements is the emergence of DSBs. Such damage usually occurs in the area of R-loops formation [18,19]. R-loops are found both in bacteria [20] and in eukaryotic cells [21]. R-loops are formed during many cellular processes, including transcription, interactions of DNA with regulatory RNAs [22], immunoglobulin class switching [23], replication of mitochondrial DNA [24], gene editing by the CRISPR systems [25,26,27], specific regulatory stages of initiation and termination of transcription [28,29] and telomere homeostasis [30]. It has been shown that R-loops accumulate predominantly in highly transcribed genomic regions, including rRNA and tRNA loci, promoter and terminator regions, and additionally, in some transposable elements, centromeres and telomeres, antisense-RNA or ncRNA genes [21,31,32,33,34]. In addition, genomic profiling of R-loops has shown that a large proportion of genes with terminal GC skew form R-loops at their 3′ ends [35].

One of the reasons for the vulnerability of the R-loop is the separation of DNA strands. ssDNA is highly accessible to metabolites, reactive oxygen (oxidative stress), DNA modifying enzymes, nucleases, or aborted activity of cellular enzymes (Figure 1) [36,37]. For example, during somatic hypermutation at the immunoglobulin locus in B-cells, activation-induced cytidine deaminase (AID) converts dC to dU in ssDNA [38] and thus makes the DNA susceptible to the base-excision-repair enzyme uracil DNA glycosylase, which in turn excises the uracil base to create an abasic site and generates a DNA lesion [39,40]. There is also a suggestion that cancer cells might be vulnerable to cytidine deaminases mutagenesis because cells accumulate an aberrant amount of R-loops [41,42]. Virus-specific RNA structures, such as dsRNA, activate the expression of nitric oxide synthase iNOS, leading to DNA damage [43,44]. ssDNA regions containing GGGG sequence may form G-quadruplex structures recognized by G4-specific endonucleases, and thus single-strand break is formed [45]. Another reason for forming the ssDNA break is topoisomerase I (Top1), which removes DNA torsional stress by nicking and resealing one strand of DNA (Figure 1). It was shown that stabilization of the Top1 cleavage complex (Top1cc) with the chemotherapeutic drug camptothecin or Top1 overexpression in yeast stimulates large-scale chromosomal rearrangements [46,47].

It is still unknown how ssDNA breaks lead to DSB formation. One hypothesis assumes that this may be the result of two spatially close ssDNA breaks. DSB occurs when adjacent single-strand breaks (SSBs) are formed on opposite strands of DNA: cleavage of R-loops by nucleotide excision repair endonucleases, including XPF, XPG, and FEN1, causes one break [48,49,50], another SSB is formed when transcription-blocking cleavage complexes of topoisomerase I (TOP1ccs) are removed by tyrosyl-DNA phosphodiesterase 1 [51] (Figure 1). Defects in transcription-blocking TOP1ccs enhance DSB formation and prevent their repair [52].

The accumulation of R-loops is associated with the disruption of the crucial factors of various biological processes, including DNA replication, transcription, cleavage, polyadenylation, and nuclear export of the mRNA [53,54,55]. The mutations in the genes of the major mammalian 3′ –> 5′ DNA exonuclease TREX1 or RNaseH2, which actively participate in the DNA replication, lead to a dramatic increase in the occurrence of R-loops [56]. TDRD3-TOP3B complex is recruited to the c-MYC gene promoter and prevents R-loop formation, likely by relaxing negative super-coiling [57]. Knockout of TDRD3, which serves as a molecular bridge for TOP3B to bind arginine-methylated histones, increases the frequency of R loops at the c-MYC locus in B-cells and induces MYC/IGH translocations [58]. The transcription-export protein complex (THO/TREX) involved in transcription elongation and mRNA export prevents R-loop accumulation [59]. Human cells infected with Kaposi’s sarcoma-associated herpesvirus express the ORF57 protein, which is involved in the processing of viral mRNA and inactivates the THO/TREX complex via binding. Consequently, it can no longer process the cellular mRNA and leads to the formation of R-loops and genome instability [54]. Top1, in addition to its DNA cleavage and relaxation activity, is also implicated in the regulation of R-loops during splicing, presumably by affecting the activity of the serine/arginine (SR)-rich family splicing factors [60]. Thus, depending on the circumstances, Top1 can either be a cause of chromosomal rearrangements as mentioned above or increase genomic stability. Another protein apparently involved in R-loop-driven DNA damage during transcriptional termination is BRCA1, which forms a complex with RNA/DNA helicase senataxin (SETX) [61].

In mammalian cells, transcription and replication are usually separated spatially and in time. Improper regulation of these processes leads to transcription—replication conflicts (TRCs) and consequently is a possible reason for developing R-loops [62,63]. Head-on collisions, where the replication fork converges with the transcription machinery, promote R-loop formation, while co-directional collisions, where the two move in the same direction, resolve R-loops [64]. Under normal conditions, RNAPII is phosphorylated at serine 2 by cyclin-dependent kinase 9 (CDK9) in complex with BET bromodomain protein 4 (BRD4). This modification appears to be important to proceed with transcription and to prevent TRC and DNA breaks [65,66]. Chromatin remodeling complexes, such as the one of Mec1, INO80 and PAF1 in budding yeast, can prevent TRC by removing RNAPII [67]; INO80 depletion in prostate cancer cells leads to increased formation of R-loops [68]. Under replicative stress, activation of the Fanconi anemia pathway results in the attraction of structure-selective endonucleases and cleavage of one of the DNA strands in the replication fork, which leads to the formation of a DSB (Figure 1) [69,70,71].

Thus, many intracellular processes that lead to the formation of R-loops eventually result in DSB. As discussed in the following chapter, RNAs also play an important role at the stage of DSB recognition and subsequent repair.

## 3. RNA Regulating the Recognition of DSB

Emerging evidence has suggested that various RNAs, especially miRNAs, regulate the initialization of DNA-damage response (DDR) (Figure 2) [72,73]. Recent studies have shown that the formation of DSB causes a shift in the expression levels of multiple miRNAs [74], although the functional significance of these changes is mostly unclear. Here, we summarize accumulated knowledge about miRNAs in DSB recognition.

At the initial stage of DDR-signaling cascade, regardless of the reparation pathway, the key role is played by PI3K-related protein kinases ataxia telangiectasia mutated (ATM), Ataxia telangiectasia and Rad3 related (ATR) and DNA-dependent protein kinase, catalytic subunit (DNA-PKcs) (Figure 2) [75,76]. In response to DNA double-strand breaks, these kinases mediate phosphorylation of histone H2AX at serine 139 (γH2AX) at the DNA damage sites [77]. γH2AX promotes the recruitment of downstream DDR-factors and amplifies the DSB signal [78]. In 2009 it was revealed that miR-24 inhibits the expression of H2AX in blood cells via binding to a highly conserved sequence in the H2AX 3′-UTR region, making cells more vulnerable to DNA damage [79]. mir-138 acts in a similar way suppressing H2AX and inhibiting γH2AX foci formation [80]. As mentioned earlier, ATM is crucial for the H2AX phosphorylation, and ATM transcripts are also targeted by several miRNAs that bind the 3′-UTR region (Figure 2). For example, ATM expression in neuroblastoma cell lines is suppressed by miR-421, which can be induced by proto-oncogene transcriptional factor N-Myc, making neuroblastoma cells more sensitive to radiotherapy [81]. In breast cancer cells, ATM expression has been linked to upregulated miR-18a, which also inhibits the formation of nuclear foci by downstream substrates H2AX and p53-binding protein 1 (53BP1) [82]. However, there is conflicting data regarding the correlation between ATM and miR-18a, presumably reflecting the complexity of ATM regulation [83]. One example of such complexity is miR-26a that suppressed ATM in glioblastoma cells leading to sensitization of the cells to radiation [84]. However, ATM downregulation by miR-26a was also able to reduce developing myocardial infarction, including ischemia-induced apoptosis and fibrosis, apparently independently of DDR [85]. Yet another miRNA affecting the level of ATM transcripts, as well as radiosensitivity of the cells, also via the 3′-UTR region, was miR-100 in a human glioma cell line M059J [86]. In addition to regulation by endogenous miRNAs, miRNA BART8-3p encoded by Epstein–Barr virus activated ATM/ATR signaling and promoted proliferation and radioresistance in nasopharyngeal carcinoma [87]. Other (PI3K)-like protein kinases that participate in cellular responses to DNA damage and coordinate H2AX phosphorylation can be regulated by miRNAs as well. For example, increased expression of miR-185 makes renal cell carcinoma cells more vulnerable to radiation by regulating ATR expression at the post-transcriptional level [88], miR-874-3p targets DNA-PKcs, which also recruits and activates the downstream components in the NHEJ pathway of DSB repair [89] and miR-101 binds to both DNA-PKcs and ATM mRNAs and regulates them post-transcriptionally in various human cell lines (Figure 2) [90].

miRNAs are not the only RNAs involved in the initial stage of DDR. For instance, it was recently shown that in MCF-7 cells, circRNA derived from SMARCA5 (circSMARCA5) could inhibit DNA damage repair function by binding to its parent gene locus, which is important for maintaining the stability of the genome [91]. Before this report, several works have demonstrated that circSMARCA5 suppresses tumor growth by acting as competing endogenous RNA for several miRNA molecules [92,93,94,95]. Another example is GAS5-derived small nucleolar RNAs that appear to have an important role in mediating the p53 response to DNA damage [96]. In addition, CU1276, a small RNA derived from tRNA, modulates DDR via repression of replication protein A1 essential for DNA repair [97,98].

When DNA break happens, transcription through the damage sites stops to prevent conflicts between transcription and repair mechanisms and to avoid genome rearrangement and tumorigenesis [99]. However, remodeled and decondensed chromatin at the damaged site resembles that at the transcriptional loci [100] and small RNAs are produced at the damaged site (Figure 3) [101]. How damage response RNAs (DDRNAs) are generated is not well defined. One of the models suggests that following a double-strand break MRN complex, which is the primary sensor of damaged DNA, participates in the recruitment of RNA Polymerase II. This induces a bidirectional transcription of damage-induced lncRNAs (dilncRNAs) [102] that form double-strand RNA by folding or pairing with already existing transcripts, followed by processing with DROSHA and DICER to generate DDRNAs (Figure 3) [103,104]. Interestingly, cells treated with enoxacin, which was found to promote DICER activity [105], show stronger DDR signaling and faster DNA repair [106].

Although DDRNAs are not directly involved in the initial recognition of DSBs and do not affect the phosphorylation of H2AX, they are crucial for proficient repair. They serve as a sequence-specific signal for downstream DDR events involving RNA-interacting enzymes. As an example, it was shown that DDRNAs are recruited by Ago2 protein, which at the same time forms a complex with Rad51, the interaction between proteins is not affected by DDRNAs. However, DDRNAs are required to guide RAD51 to single-strand DNA filaments and to promote subsequent homologous reparation [101].

DDRNAs can also mediate chromatin remodeling by acting as guiding molecules for the recruitment of chromatin-modifying enzymes MMSET and Tip60 to DSBs via Ago2. This process leads to chromatin relaxation, increasing the accessibility of the break site for Rad51 and BRCA1, promoting homology-directed repair [107]. 53BP1 is also associated with DDRNAs and dilncRNAs engaged in the mechanisms of 53BP1 focus formation [102]. Another function of DDRNAs is that they can target potentially damaged mRNA in the proximity of DSB [108]. Thus, damage response RNAs may play various functional roles in the DNA damage response.

## 4. All Pathways of Homology-Directed Repair Are Controlled by RNAs

Homology-directed repair (HDR) is preferable because it is more likely to preserve the original DNA sequence than nonhomologous end-joining (NHEJ). The choice of the repair pathway depends on the availability and activity of key factors of these mechanisms and the availability of a homologous matrix for the damaged sequence. Contrary to NHEJ, which uses limited (0–4 bp) base pairing of processed ends, homology-mediated repair pathways require extensive base pairing between the repaired strand and a template. Depending on the pathway, HDR uses homologous sequences of 2–20 nucleotides for microhomology-mediated end joining (MMEJ or altEJ), about 25 nucleotides or longer for single-strand annealing pathway (SSA), or even more than a hundred nucleotides of base pairing for more conservative synthesis-dependent strand annealing (SDSA) and classical DSB repair homologous recombination pathway (cHR) [109]. Execution of cHR, SDSA and SSA can be divided into four main steps: (1) resection of 5′ ends, (2) displacement of RPA with RAD51 and subsequent 3′ strand invasion, (3) nascent strand synthesis, (4) ligation. Resection of 5′ ends is started by MRN complex (MRE11, RAD50, NBS1 proteins), which is activated by CtIP tetramer (RBBP8), and then the resection is extended by EXO1 and BLM-DNA2 proteins (Figure 4) [110]. Recently, RNA polymerase III was shown to protect 3′ overhangs by forming RNA-DNA hybrids [111]. RPA is displaced by BRCA2 and BRCA1-BARD1 with the help of PALB2, and 3′ end invasion is facilitated by RAD52 (Figure 4) [112,113]. cHR, SDSA and SSA predominantly take place in the G2 and S phases of the cell cycle. When sister chromatids are available, expression of RAD51 and RAD52 is higher, and phospho-regulated repair proteins are in an active state [114,115,116]. MMEJ is also more prevalent in G2 and S, but this restriction is less pronounced because MMEJ uses much shorter 5′ resection [117]. MMEJ is impaired without polymerase POLQ, which extends paired 3′ ends, and NAD+ ADP-ribosyltransferase 1 (PARP1), which is suggested to recruit POLQ [118]. MMEJ is considered less frequent than other types of DNA repair. However, if NHEJ is suppressed, it could become prevalent in the G1 phase of the cell cycle [119]. The choice of the HDR pathway is based on the presence of employed proteins and the availability of a template [120]. Whenever one pathway is stalled, the likelihood of alternatives increases.

All homology-directed repair pathways can be controlled by RNAs, either by pairing with single-strand DNA regions or by influencing the participating proteins. The latter is by far the most studied model of RNA participation in HDR. Multiple long noncoding RNAs are employed in the regulation of expression of HDR factors (Figure 4). One of the most notable lncRNAs, which influences homologous recombination is PCAT1. It was initially identified as an outlier in prostate cancer tumors [121]. In the LNCaP prostate cancer cell line, PCAT1 knockdown led to an increased BRCA2 protein expression, while PCAT1 overexpression decreased BRCA2 protein level and impaired repair of double-strand brakes, thus sensitizing cells to PARP1 inhibitors and radiation [122]. High levels of PCAT1 correlated with poor prognosis in multiple cancer types [123,124,125,126]. Lnc-RI (radiation-induced) acts as a competitive endogenous RNA for RAD51 by sponging miR-193a-3p. Knockdown of lnc-RI leads to a decrease in mRNA and protein levels of RAD51 in U2OS and HeLa cells. It was also shown that micronucleus frequency in peripheral blood lymphocytes of healthy donors is negatively correlated with lnc-RI expression [127]. Another example of lncRNA, which regulates homology-mediated repair genes, is NEAT1. It is an essential structural element of regulatory paraspeckle nuclear bodies [128]. NEAT1 knockdown with LNA-gapmeR antisense oligonucleotide downregulates genes involved in DNA repair processes, including the homologous recombination pathway, which in turn results in massive DNA damage [129].

Besides regulation of expression by lncRNAs, each repair factor has a number of binding miRNAs confirmed with various experimental evidence [130]. Here we focus on miRNAs, which not only influence the expression of repair genes but also have been shown to affect DNA repair efficiency (Figure 4). As an example, expression of miR-223 exhibited a correlation with cancer progression in Barrett’s esophagus-associated esophageal adenocarcinoma. miR-223 decreases the expression of the PARP1 gene, thus sensitizing adenocarcinoma cells to chemotherapy [131]. Another miRNA that was shown to downregulate PARP1 expression is miR-7-5p. The reduction of PARP1 expression by miR-7-5p inhibits DNA repair and induced cisplatin resistance in cervical cancer cells [132]. miR-1255b, miR-193b and miR-148b were shown to regulate the expression of BRCA1, BRCA2, and RAD51 proteins leading to enhanced PARP inhibitor sensitivity [14]. Polymorphisms that enhance the effect of miRNAs on the stability of RAD52 mRNA are associated with an increased risk of HBV-related hepatocellular carcinoma [133].

Another way for lncRNAs to affect HDR proteins (apart from regulating their expression level) directly binds to these factors. HIF-1α inhibitor at translation level lncRNA (HITT) provides an example of such activity. HITT binds the HEAT repeat domain of ATM, thus blocking ATM recruitment through the MRE11-RAD50-NBS1 complex (Figure 4). HITT is downregulated in multiple types of cancer but can be elevated in response to double-strand breaks through the activation of EGR1, resulting in decreased ATM activity [134].

One more mode of lncRNA influence on HDR so far has been studied only in yeasts. In fission yeast Schizosaccharomyces pombe, it was shown that meiosis-specific lncRNA sme2 accumulates in its locus and thereby facilitates pairing of homologous loci [5]. Deletion of sme2 decreases this pairing; transposition to other chromosomal sites directs pairing to these regions.

## 5. Different Types of RNAs Are Involved in All Stages of Nonhomologous End-Joining

When the activity of factors involved in homology-mediated repair is reduced, DSBs are eliminated by the nonhomologous end-joining (NHEJ) mechanism. It is an error-prone DSBs repair mechanism involving several stages. Upon recognition of the break, several protein complexes attached to its ends, including ATM and 53BP1. Next, the Ku70/Ku80 heterodimer and DNA-PKcs are recruited to the damaged region, activating several proteins involved in eliminating the gap, in particular Artemis nuclease and other nucleases that cleave protruding single-strand DNA regions. Finally, DNA ligase IV and its cofactor XRCC4 link the free ends of the DNA, eliminating the break (Figure 5) [135,136,137,138].

The main advantage of NHEJ is its versatility. This mechanism does not require a homologous matrix and functions during the G0/G1/S/G2 phases of the cell cycle, although it is most active in the G1 phase [139]. NHEJ also plays a critical role in the process of V(D)J recombination that generates B-cell and T-cell receptor diversity in the vertebrate immune system [140]. However, its drawbacks are lower repair fidelity due to possible shortening or lengthening of the ends and the possibility of connecting nonhomologous DNA sites [141]. Moreover, this mechanism is the main DSBs repair method in cancer cells, crucially contributing to their resistance to radiation and chemotherapy [142,143].

A number of RNAs are involved in the control and organization of NHEJ (Figure 5). In particular, there are many examples of RNAs controlling the expression of factors involved in the NHEJ mechanism, for instance, miR-101 and miR-874-3p target DNA-PKcs (as mentioned above). miR-124, miR-502 and miR-545 regulate Ku70 [144,145,146], miR-526b, miR-622 and miR-218-5p control the level of Ku80 [89,147,148], miR-34a suppresses the expression of 53BP1 [149]. For some factors, in addition to miRNA, competing endogenous RNAs (ceRNA) have been described that are able to counteract the effects of specific miRNAs by direct binding (Figure 5) [150]; for example, miR-4727–5p was shown to inhibit the expression of DNA ligase IV, while aforementioned ceRNA lnc-RI, reversed this effect [151]. In addition, for miR-151a-3p, which inhibits XRCC4 expression, it was shown that ceRNA SBF2-AS1 neutralizes the effect of this miRNA [152]. lncRNAs can also control transcription by interacting with transcription and RNA processing factors. For example, lncRNA LIRR1 suppresses the expression of Ku70, Ku80 and RAD50 by activating p53 [153]. It should be noted that these articles indicate the effect of RNA on the expression of various repair factors, but their impact on the process of reparation itself is not always proven.

Many proteins involved in the repair of double-strand breaks are known to have RNA-binding capacities [4]. Thus, different types of RNA cannot only participate in the regulation of the expression of repair factors but also be directly involved in this process. For example, lncRNA LINP1 was shown to participate in the assembly of a complex that combines such key factors of the NHEJ mechanism as Ku80 and DNA-PKcs (Figure 5). Suppression of LINP1 inhibited the processes of double-strand break repair and led to an increase in the sensitivity of cells to radiation [154].

As it was mentioned before, the NHEJ mechanism is the main way to repair double-strand breaks in non-dividing cells and cells in the G1 phase, when transcribed DNA regions are most vulnerable to damaging effects due to the formation of R-loops. Therefore, it is not surprising that genomic localization of the NHEJ factors correlates with transcription. Some results indicate that the RNA of the gene in which the break occurred can bind to the free ends of the DNA. Thus, the NHEJ mechanism could employ nascent RNA, the synthesis, which was the initial cause of the DBS formation. In this case, RNA not only promotes the spatial proximity of the ends of DNA but also prevents the coupling of parts of different genes [155]. Another study found a key role of RNA polymerase II in the gap elimination by the NHEJ mechanism via the formation of a complex with RBM14 and Ku factors that can trigger RNA synthesis near the gap region even if it occurred in a transcriptionally inactive region. Apparently, the role of such RNAs at the region of rupture is to attract and activate proteins involved in connecting the ends of DNA, such as XRCC4 [156,157]. Thus, the synthesis of RNA in the area of the break can be important not only at the stage of recognition but also at the stage of repair of DSB.

## 6. Noncoding RNAs Directly Provoke Chromosomal Rearrangements

The cases were described where RNA directly guides chromosomal rearrangements. Currently, the role of RNAs in genome rearrangements has been mostly studied in ciliates. These ciliate unicellular organisms have two types of nuclei: micronuclei are involved only in sexual reproduction, and transcriptionally active macronuclei, which participate in the regulation of vital functions of the organism. The ciliate *Oxytricha trifallax* forms new macronuclei from the micronuclei in the process of conjugation when 90% of the micronuclei genome is removed (Figure 6). Deleted sequences are termed internally eliminated sequences (IES). These sequences contain transposons and satellite DNA. Macronuclear destined sequences (MDS) are merged, forming new combinations of genes [158]. This complex process has been shown to be controlled by small noncoding piRNAs (piwi-interacting RNA) and long noncoding RNAs, which are produced in the macronucleus before conjugation begins. During the formation of the new macronucleus, a precursor piRNA is synthesized from the genome of the parental macronucleus, which is processed to a 27 nt piRNA. These piRNAs in complex with PIWI family proteins (namely Otiwi1) mark MDS sites in the developing macronucleus and thus prevent their removal (Figure 6) [7]. In addition, lncRNAs are synthesized from the parental macronucleus genome that program the order and orientation of the MDS segments in the developing macronucleus. Microinjection of alternative DNA or RNA resulted in an inherited alternative genome reorganization pathway [159].

In the ciliates Tetrahymena and *Paramecium*, genome rearrangement removes about 30% of the micronucleus genome, and RNAs are also actively involved [8,160]. In this case, special small dsRNAs, so-called scanning RNAs (scanRNAs), act as markers (Figure 6) [161,162]. These RNAs are formed during the early stages of conjugation, processed by Dicer-like proteins Dcl2/3 in *Paramecium* or Dcl1p in Tetrahymena, and then transferred to the parental macronucleus as part of the complex with PIWI proteins. In the parental macronucleus, the entire genome is transcribed, and the scanRNAs bind complementarily to the macronucleus RNA. In this process, most of the RNAs of the old macronucleus degrade (along with the corresponding scanRNAs), except for the IES sequences, which are absent from the old macronucleus but are present as part of the scanRNA-protein complexes. The remaining complexes go into the developing new macronucleus, and there serve as a template for labeling regions of the genome that are subsequently deleted [163].

Thus, all representatives of ciliates employ RNAs as crucial elements of genome rearrangement. Interestingly, other genera of ciliates do so in significantly different ways. In addition to differences in function, RNAs of distinct ciliates have structural differences. *Paramecium* scanRNAs have a 5′-UNG signature, and scanRNAs without 5′-U contain a complimentary CNA motif 2 nt from the 3′-end, indicating that scanRNAs form duplexes with 3′ overhangs and are processed from dsRNA structures by Dicer proteins [161]. This CNA signature is absent in Oxytricha, and it is unclear whether Oxytricha piRNAs originate from ssRNAs or from dsRNAs. In addition, *Tetrahymena* scanRNAs contain 3′-terminal methylation [164], whereas Oxytricha piRNAs lack 3′-terminal modifications [7].

Mammals also produce piRNAs, which function as transposon-inhibiting factors via transposon gene silencing, emphasizing their role as “genome trash selectors”. However, in this case, RNAs do not direct chromosomal rearrangements but rather suppress them by interfering with mobile genetic elements [165]. On the other hand, trans-splicing that has been observed in mammals may allow them to recombine different genes at the RNA level [166]. One intriguing finding was that trans-splicing in healthy human endometrial cells produced chimeric mRNA JAZF1-JJAZ1. This chimeric RNA also has been observed in human endometrial stromal sarcomas cells with chromosomal rearrangements between the JAZF1 and JJAZ1 genes [167]. These data suggest that such hybrid RNA may act as a catalyst that ensures convergence of two genomic loci and promotes chromosomal rearrangements between the corresponding genes [168]. Recently, the ability of hybrid RNAs to stimulate TMPRSS2-ERG and TMPRSS2-ETV1 chromosomal rearrangements was studied. In the LNCaP prostate adenocarcinoma cell line, which initially lacks these rearrangements, the introduction of artificial fused antisense RNA sequences of TMPRSS2 and ETV1 (or TMPRSS2 and ERG) genes resulted in the appearance of the corresponding rearrangements, while the sense hybrid RNAs had no effect. Overexpression of an endogenous RNA AZI1 that possesses complementary regions to the TMPRSS2 and ERG genes near the translocation area also led to the appearance of the corresponding translocation. The proposed mechanism of RNA-mediated chromosomal rearrangement involves stabilization of the interaction between free DNA ends by a specific endogenous RNA that directs DSB repair, leading to the formation of the rearrangement, although the suggested mechanism is yet to be confirmed by direct experiments [6].

## 7. Future Prospects

The widespread availability of high-throughput sequencing has increased the scientific community’s understanding of the importance and diversity of RNA molecules in eukaryotic cells. Genome-wide analysis of RNA–RNA [29130190, 32320281], RNA-DNA [30700807] and RNA–protein connections [22213601, 29400715, 32140725] provides new RNA targets for further detailed research. However, current genome-wide data contain a substantial amount of noise, limiting its applicability mostly to highly expressed genes. With the improved quality of large-scale predictions, the number of emerging investigations and valuable results will continue to increase. The function of most noncoding RNAs has not yet been discovered, but it is already clear that various types of noncoding RNAs can participate, among others, in processes of maintaining genome stability (32764716). Data on the role of various RNAs in DNA repair brought insights about oncogenesis processes, allowed to identify the causes of individual susceptibility to various cancers and improve understanding of the tumor cell resistance to chemo- and radiotherapy. Methods are being actively developed for the targeted delivery of short RNA and DNA to destroy certain RNAs in the cell (28423924; 31775591). In addition, approaches to human cell genome editing continue to improve (32051598). Using these technologies, it may be possible to inhibit the expression of RNAs involved in tumor cell resistance to therapy.

## 8. Conclusions

As discussed in this review, RNA is implicated in DSB repair in many ways and literally at every stage of this process. It may be interesting to look at this intimate involvement from the evolutionary perspective. Indeed, according to the “RNA world” hypothesis, during abiogenesis, RNA both stored genetic information and performed the central role in its implementation [169]. Therefore, the existence and reproduction of long RNAs would have required mechanisms to restore RNA after damage [170]. In the modern world, functional RNA genome repair mechanisms can be observed in influenza viruses, polioviruses, and coronaviruses [171,172,173]. The elimination of damage in these RNA-containing viruses mostly occurs by the “copy-choice” recombination method using homologous RNA as an auxiliary matrix. The human immunodeficiency virus uses a similar mechanism to create a full-fledged DNA copy of the genome from several RNA matrices [174]. One can hypothesize that as the role of the main genetic information repository transferred from RNA to DNA, repair mechanisms of RNA genomes were adapted for DNA repair, explaining the key role of RNA molecules in the control of genome stability in modern organisms. From the evolutionary point of view, the control of genome stability plays a dual role: on one hand, chromosomal rearrangements can reduce the viability and fertility of organisms; on the other hand, genomic duplications are of the utmost importance for the emergence of new genes [175]. As described in the previous chapter, the processes of controlled chromosome rearrangement in unicellular organisms occur with specific RNAs. Moreover, RNAs that guide certain genome rearrangements have been found in human cells as well. Further study of RNAs that stimulate certain chromosomal aberrations may help us better understand both the evolution of species and the evolution of individual cells in the process of carcinogenesis.

## Figures and Tables

**Figure 1 biomolecules-11-00550-f001:**
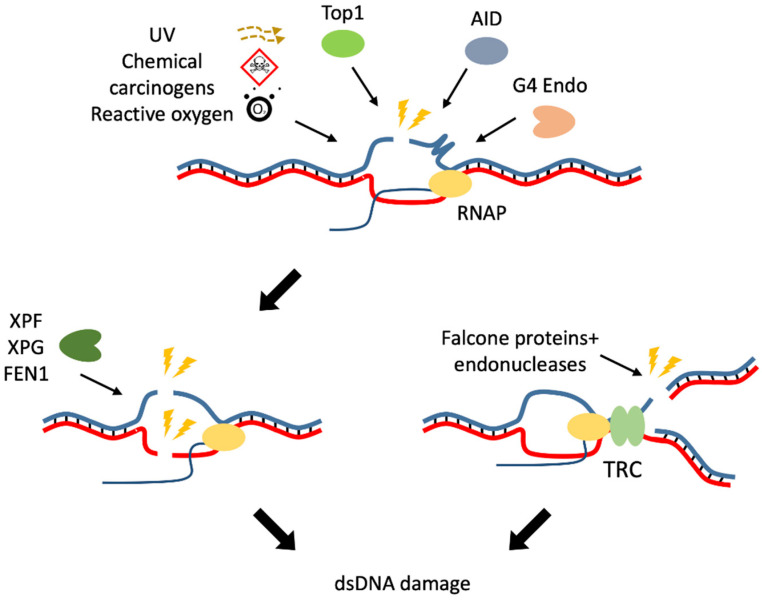
Origin of double-strand breaks under the influence of R-loops. The causes of single-strand breaks (in the upper part of the figure) may be exogenous, such as ultraviolet radiation (UV), chemical carcinogens, reactive oxygen, or might be endogenous, such as the activity of topoisomerase I (Top1), activation-induced cytidine deaminase (AID) or G4-specific endonucleases (G4 Endo). Another single-strand break can occur in the same region of DNA due to endonucleases xeroderma pigmentosum group G and F (XPG and XPF) or flap endonuclease 1 (FEN1), leading to double-strand DNA damage (dsDNA damage). Transcription–replication conflict (TRC) may also cause dsDNA damage (in the lower right part of the figure).

**Figure 2 biomolecules-11-00550-f002:**
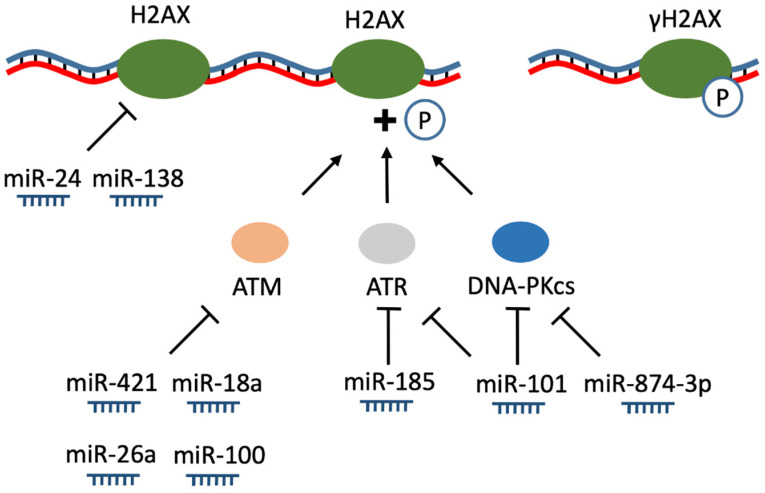
miRNA regulation of DNA-damage recognition. miRNAs affect the process of histone phosphorylation controlling both the mRNA level of the H2A histone family member X (H2AX) itself and expression of protein kinases: phosphoinositide 3-kinase (PI3K)-related ataxia telangiectasia mutated protein kinase (ATM), ataxia telangiectasia and Rad3 related protein kinase (ATR) and DNA-dependent protein kinase, catalytic subunit (DNA-PKcs). MiR-24 and mir-138 suppress H2AX expression and inhibiting γH2AX foci formation. Synthesis of ATM is suppressed by miR-18a, miR-421, miR-26a and miR-100. miR-185 targets ATR, miR-874-3p inhibits expression of DNA-PKcs, miR-101 binds to both DNA-PKcs and ATM mRNAs.

**Figure 3 biomolecules-11-00550-f003:**
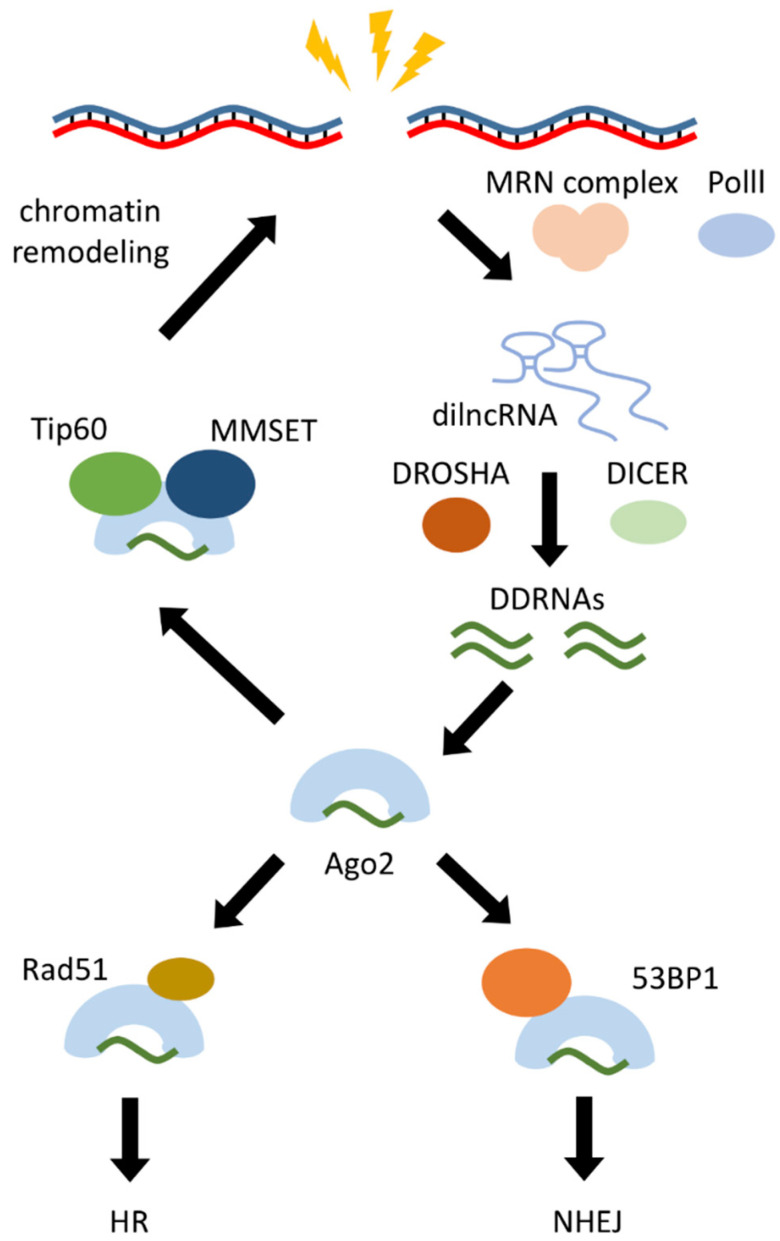
Synthesis and function of damage response RNAs (DDRNAs). Damage-induced lncRNAs (dilncRNA) is a precursor of DDRNA that is synthesized at the site of rupture by polymerase II (PolII) attracted by the MRN protein complex consisting of Mre11, Rad50 and Nbs1. They are processed by DROSHA and DICER to DDRNAs and participate in chromatin remodeling by using Tip60 and multiple myeloma SET domain (MMSET) and regulation of nonhomologous end joining (NHEJ) and homologous recombination (HR) using p53-binding protein (53BP1) and Rad51.

**Figure 4 biomolecules-11-00550-f004:**
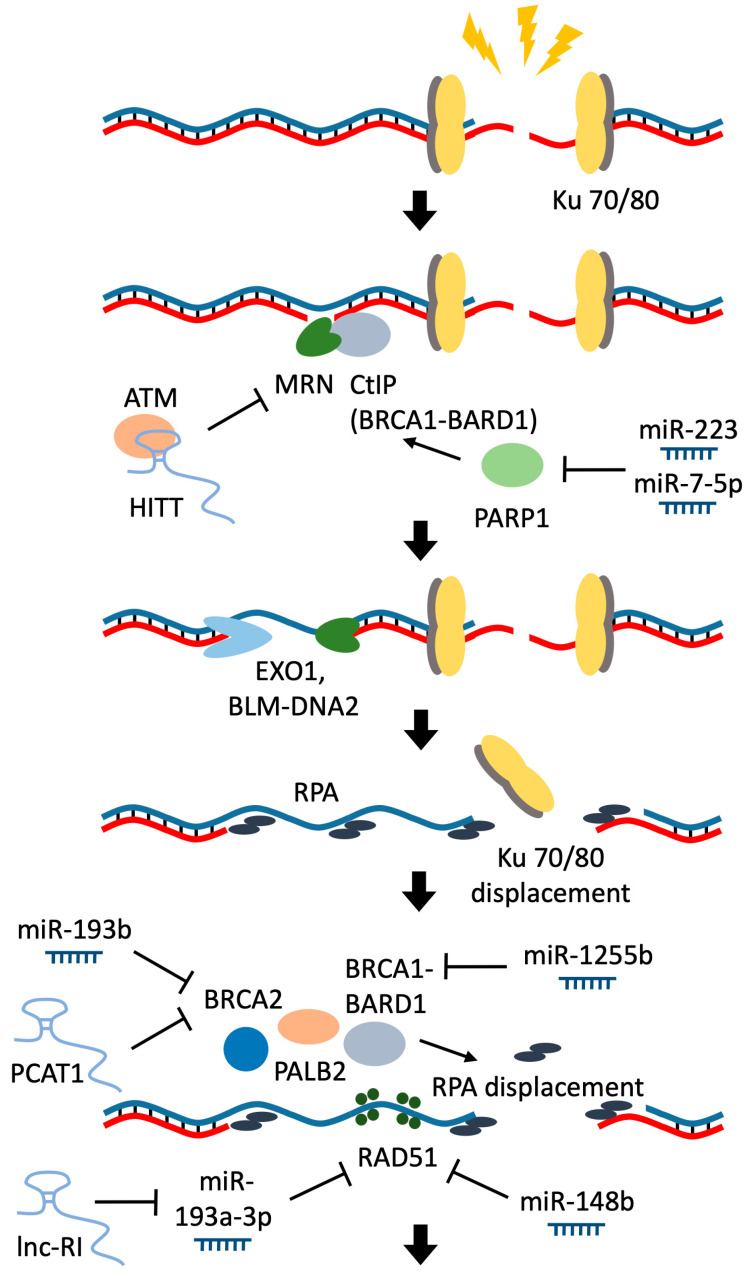
RNAs that regulate homology-directed repair (HDR) during the resection of 5′-ends and replication protein A (RPA) displacement. LncRNA HITT disrupts the ability of ataxia telangiectasia mutated protein kinase (ATM) to bind with the MRN complex, leading to reduced HDR. MiR-7-5p and miR-223 decreased the expression of the poly-(ADP-ribose) polymerase (PARP1). PARP1 through the breast cancer type-1 susceptibility protein (BRCA1) and BRCA1 associated RING domain 1 (BARD1) complex controls the work of the CtIP endonuclease. After activation of this nuclease, the resection is expanded by exonuclease 1 (EXO 1) with the help of bloom syndrome helicase (BLM) and DNA Replication helicase 2 (DNA2). RPA binds to single-strand DNA, and it is displaced by BRCA2 and BRCA1-BARD1 with the help of Partner and localizer of BRCA2 (PALB2). Expression of BRCA1 and BRCA2 is controlled by miR-1255b and miR-193b, respectively. Expression of BRCA2 is also suppressed by lncRNA prostate cancer-associated transcript 1 (PCAT1). Protein RAD51 is necessary for 3′ strand invasion and passing further stages of HDR. Expression of RAD51 is controlled by miR-148b and miR-193a-3p. The activity of the last miRNA is suppressed by radiation-induced lncRNA (Lnc-RI).

**Figure 5 biomolecules-11-00550-f005:**
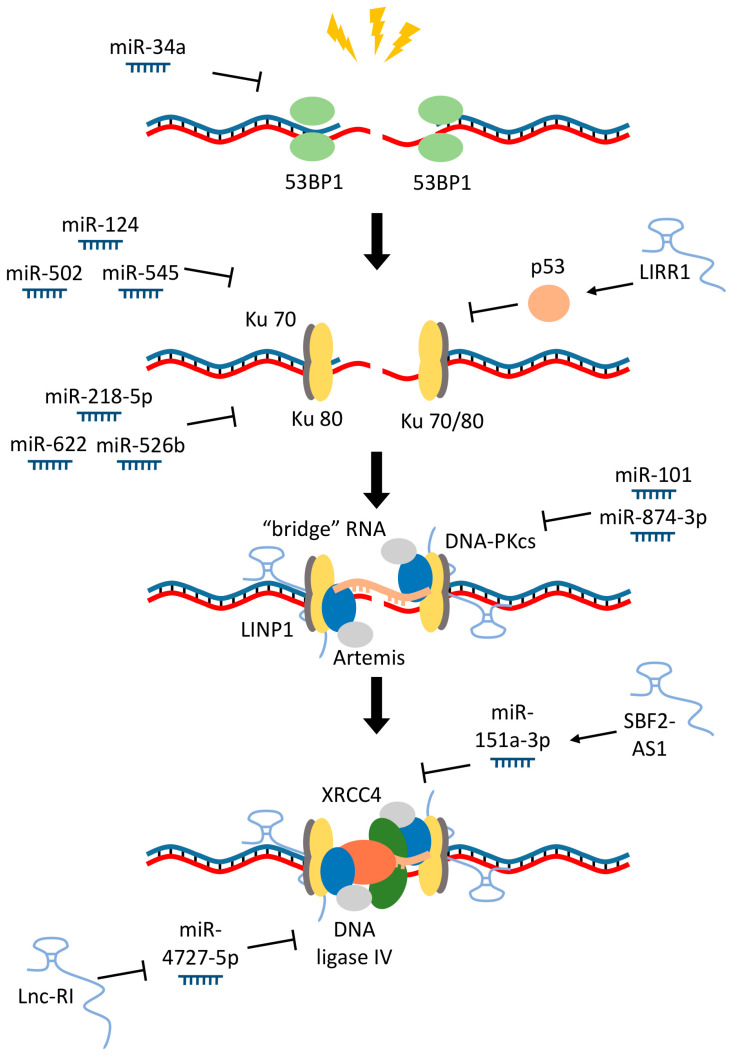
RNAs that regulate nonhomologous end-joining (NHEJ). At the first stage of NHEJ, the p53-binding protein 1 (53BP1) binds to the break region. The synthesis of the protein is controlled by miR-34a. Next, the Ku70/Ku80 heterodimer and DNA-dependent protein kinases (DNA-PKcs) are recruited to the damaged region. The stability of these three factors mRNA is controlled by the following miRNA: miR-124, miR-502 and miR-545 (Ku70); miR-526b, miR-622 and miR-218–5p (Ku80); miR-101 and miR-874-3p (DNA-PKcs). Furthermore, long intergenic radiation-responsive ncRNA1 (LIRR1) suppresses the expression of Ku70 and Ku80 by activating p53. lncRNA in nonhomologous end-joining pathway 1 (LINP1) to participate in the assembly of a complex with Ku80 and DNA-PKcs. “Bridge” RNA could hold together the DNA strands in the break area. At the last stage, DNA ligase IV and its cofactor X-ray repair cross-complementing protein 4 (XRCC4) link the free ends of the DNA, eliminating the break. Expression of DNA ligase IV and XRCC4 is controlled by miR-4727-5p and miR-151a-3p correspondingly. These miRNAs are blocked by lncRNA radiation-induced (lnc-RI) and SBF2 Antisense RNA 1 (SBF2-AS1).

**Figure 6 biomolecules-11-00550-f006:**
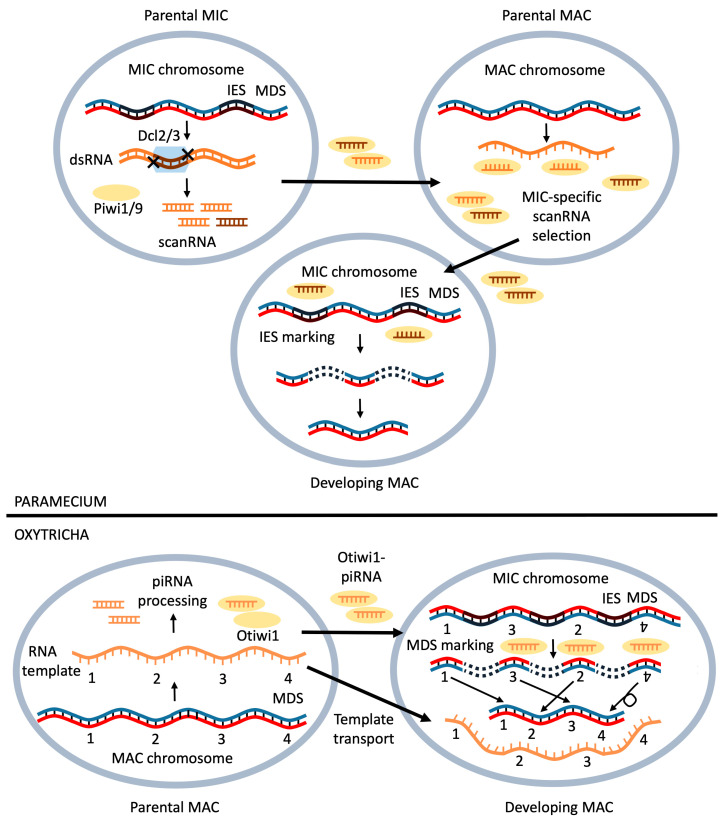
The participation of RNAs in the formation of a macronucleus (MAC) from a micronucleus (MIC) in ciliates Tetrahymena, Paramecium and *Oxytricha trifallax*. The upper part of the figure illustrates the role of scanning RNAs (scanRNAs) in *Tetrahymena* and *Paramecium*. MIC genome produces double-stranded RNA (dsRNA), which is processed into scanRNAs by Dicer-like proteins Dcl2/3. Piwi1/2 forms a complex with scanRNA, and together, they migrate into parental MAC. The scanRNAs bind complementarily to the macronucleus RNA, and after MIC-specific selection, only internally eliminated sequences (IES) are transported into developing MAC. Bellow, it is shown how RNAs direct genome rearrangement in *Oxytricha trifallax*. Otiwi1 binds to piwi-interacting RNA (piRNA), and Otiwi1–piRNA complex is transported into developing MAC. The Otiwi1-piRNA complex recognizes and marks macronuclear destined sequences (MDS) regions. RNA template helps to guide rearrangements. In the lower right illustration, segments 2 and 3 are switched, and segment 4 is inverted.

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
