# Peer review of "The Role of RNA in DNA Breaks, Repair and Chromosomal Rearrangements"

_biomolecules, 2021, doi:10.3390/biom11040550_

Round 1
Reviewer 1 Report
The author drafted the review titled The role of RNA in DNA breaks, repair and chromosomal rearrangements, this is a big topic and a great interesting problem.
Noncoding RNAs in DNA repair is very important and already have many related reviews. In addition to growing evidence for their multifunctional role in diverse biological processes a connection between ncRNAs and DNA repair in genome safeguarding has been recently described.
Major issues:
Due to miRNAs are not the only RNAs involved in the initial stage of DDR. Another reason is the review title is The role of RNA in DNA breaks, repair and chromosomal rearrangements.
It was reported that circRNAs could be used as biomarkers to distinguish PML–RARA-driven APL with other subtypes of AML (You and Conrad, 2016; Li et al., 2018a; Zhang et al., 2014, 2016). So, I suggestion the author need summary a little bit about tRNAs and snoRNAs and circRNAs involved in the initial stage of DDR.
And the author talked more about the role of RNA in DNA breaks, repair and chromosomal rearrangements in the process of carcinogenesis, I suggestion the author also need talk about virus–host interactions related in the conclusions part according to the title. Compared to the recognition of pri-miRNA substrates by Microprocessor, the distinction between self and non-self RNAs in the cell is much more complex. Endogenous RNAs may come in different forms with a variety of end chemistry, modification status, and associated protein repertoire. Then will bring new insights into autoimmunity, virus–host interactions, and RNA therapeutics to the readers.
Minor issues:
In line 205, damage response RNAs (DDRNAs) or damage-induced RNAs, I suggest the author select use damage response RNAs.
In line 204, Another function of DDRNAs RNAs is that they can target potentially damaged mRNA, here delete RNAs.
In line 269, miRNAs miR- 1255b, miR-193b and miR-148b, here should delete miRNAs.
I feel the overall manuscript has potential and has interesting for the readers, I would be happy to review the revised manuscript.
Author Response
Thank you for your valuable comments and suggestions. They certainly helped us to improve the manuscript. We hope that we managed to take into account all your tips.
1) Due to miRNAs are not the only RNAs involved in the initial stage of DDR. Another reason is the review title is The role of RNA in DNA breaks, repair and chromosomal rearrangements.
It was reported that circRNAs could be used as biomarkers to distinguish PML–RARA-driven APL with other subtypes of AML (You and Conrad, 2016; Li et al., 2018a; Zhang et al., 2014, 2016). So, I suggestion the author need summary a little bit about tRNAs and snoRNAs and circRNAs involved in the initial stage of DDR.
We’ve expanded the section of review devoted to the recognition of DNA damage with a description of examples of circular RNAs, RNA derived from tRNA and snoRNA involved in this process (lines 201-210).
2) And the author talked more about the role of RNA in DNA breaks, repair and chromosomal rearrangements in the process of carcinogenesis, I suggest the author also needs to talk about the virus–host interactions related in the conclusions part according to the title. Compared to the recognition of pri-miRNA substrates by Microprocessor, the distinction between self and non-self RNAs in the cell is much more complex. Endogenous RNAs may come in different forms with a variety of end chemistry, modification status, and associated protein repertoire. Then will bring new insights into autoimmunity, virus–host interactions, and RNA therapeutics to the readers.
The information about the role of viruses and virus-host interaction in DNA repair and chromosomal rearrangements was added to multiple sections of the review. This topic is declared in the introduction (lines 51-55), the influence of virus-specific RNA in DNA damage is mention in the formation of double-strand breaks section (lines 82-83). The effect of viral miRNA is described in recognition of DSB section of the manuscript (lines 190-192). The association of polymorphisms with HBV-related hepatocellular carcinoma was added to homology-directed repair section (lines 331-332).
3) In line 205, damage response RNAs (DDRNAs) or damage-induced RNAs, I suggest the author select use damage response RNAs.
We replaced damaged-induced RNA with damage response RNAs (line 251).
4) In line 204, Another function of DDRNAs RNAs is that they can target potentially damaged mRNA, here delete RNAs.
The redundant RNAs mention was deleted (line 250).
5) In line 269, miRNAs miR- 1255b, miR-193b and miR-148b, here should delete miRNAs.
Excess word miRNAs has been deleted (line 329).
Reviewer 2 Report
The manuscript entitled ‘The role of RNA in DNA breaks, repair and chromosomal rearrangements’ by Murashko et al represents a very interesting review where the authors presented an overview of the role of different types of RNA in the regulation of the mechanisms of DSB formation, recognition and reparation, deviation in which may lead to a consequent chromosomal rearrangement formation. The images presented are very good and improved the quality of the review. However interesting, I considered that is a narrative review of the literature.
In overall, I consider that the premise of this study is very interesting and important for the field and I will perform some comments and suggestions.
Minor concerns:
- The lack of critical analysis is the major drawback of this paper. The authors have provided a list of several studies, but a critical analysis was rarely reported.
- A section of future perspectives is also missing; however, some information is included in the conclusion section, but they sould be changed for this new section.
- One interesting point indicated in the review objectives (lines 54-55, referent to ‘discussion the significance of these RNAs for the process of oncogenesis’ was less explored.
- The figures legend should be more descriptive, and a list of abbreviations included.
- The figures are appearing before the first citation in the text; they should be stated in brackets, for example (figure 1) and cited all over the correspondent text.
- Lines 30-31, the statement ‘The formation of chromosomal rearrangements is also linked to the activity of a number of RNAs’ should end with one or more references.
- Some verbs forms should be revised all over the manuscript (example: lines 59,60)
- The miR-5a is reported in Figure 2, however is not described in text.
- The figure 6 is not readable. Please improve this image quality!
Author Response
Thank you for your valuable comments and suggestions. They certainly helped us to improve the manuscript. We hope that we managed to take into account all your tips.
1. The lack of critical analysis is the major drawback of this paper. The authors have provided a list of several studies, but a critical analysis was rarely reported.
According to your advice, additional criticism of listed articles was added to the review. The drawback of the cited article (Gupta et al 2018) was added to the section devoted to the direct participation of noncoding RNAs in chromosomal rearrangements (lines 496-497). In the NHEJ section of the manuscript, we’ve discussed lacking data on RNA functions (lines 393-395). Also, the summary of controversial Top1 activity was added to the section about DSB formation (lines 125-127).
2. A section of future perspectives is also missing; however, some information is included in the conclusion section, but they should be changed for this new section.
Following your recommendations, we’ve added a new section about future prospects of this field to the review (lines 499-517).
3. One interesting point indicated in the review objectives (lines 54-55, referent to ‘discussion the significance of these RNAs for the process of oncogenesis’ was less explored.
We’ve expanded the discussion of the roles of RNAs in oncogenesis in several sections of the review. The role of viral RNAs in oncogenesis is described in the introduction (lines 51-55). The effect of viral miRNA on nasopharyngeal carcinoma is now mentioned in recognition of DSB section of the manuscript (lines 190-192). Also, this section was expanded with a description of circSMARCA5 influence on tumor growth (lines 201-206). In the HDR section of the review, we’ve added information about an increased risk of hepatocellular carcinoma associated with RAD52 miRNAs (lines 331-332).
4. The figures legend should be more descriptive, and a list of abbreviations included.
Figures' legends were significantly expanded. All abbreviations were explained.
5. The figures are appearing before the first citation in the text; they should be stated in brackets, for example (figure 1) and cited all over the correspondent text.
Additional references to the figures were added to the text. All of them are now enclosed in brackets.
6. Lines 30-31, the statement ‘The formation of chromosomal rearrangements is also linked to the activity of a number of RNAs’ should end with one or more references.
Four references were added to this phrase (line 30-31).
7. Some verbs forms should be revised all over the manuscript (example: lines 59,60)
Some mistakes in the verb forms have been found and corrected (lines 64, 116, 117, 328, 461-462).
8. The miR-5a is reported in Figure 2, however is not described in text.
miR-5a was added to the figure by mistake. This error was corrected and miR-5a was substituted with miR-18a mentioned in the text.
9. The figure 6 is not readable. Please improve this image quality!
The composition of figure 6 was changed and the quality was improved.